# High Sensitive Biosensors Based on the Coupling Between Surface Plasmon Polaritons on Titanium Nitride and a Planar Waveguide Mode

**DOI:** 10.3390/s20061784

**Published:** 2020-03-23

**Authors:** Junior Asencios, Ramiro Moro, Clemente Luyo, Arturo Talledo

**Affiliations:** 1Facultad de Ciencias, Universidad Nacional de Ingeniería, Lima 15333, Peru; jr.david.ar@gmail.com (J.A.); cluyo@uni.edu.pe (C.L.); 2Tianjin International Center for Nano Particles and Nano Systems, Tianjin University, Tianjin 300072, China; ramiroalfredomoro@gmail.com

**Keywords:** surface plasmon polaritons, plane waveguide, Fano resonance

## Abstract

High sensitivity biosensors based on the coupling of surface plasmon polaritons on titanium nitride (TiN) and a planar waveguide mode were built; they were proved by sensing three different media: air, water and dried egg white; sensors described here could be useful for sensing materials with a refractive index between 1.0 and 1.6; in particular, materials of biological interest with a refractive index in the range 1.3–1.6, like those containing biotin and/or streptavidin. They were built by depositing Nb_2_O_5_/SiO_2_/TiN multilayer structures on the flat surface of D-shaped sapphire prisms by using the dc magnetron sputtering technique. Attenuated total reflection (ATR) experiments in the Kretschmann configuration were accomplished for the air/TiN/Prism and S/Nb_2_O_5_/SiO_2_/TiN/Prism structures, S being the sample or sensing medium. ATR spectra for plasmons at the TiN/air interface showed a broad absorption band for angles of incidence between 36 and 85°, with full width at half maximum (FWHM) of approximately 40°. For the S/Nb_2_O_5_/SiO_2_/TiN/Prism structures, ATR spectra showed a sharp reflectivity peak, within the broad plasmonic absorption band, which was associated with Fano resonances. The angular position and FWHM of the Fano resonances strongly depend on the refractive index of the sensing medium. ATR spectra were fitted by using the transfer-matrix method. Additionally, we found that angular sensitivity and figure of merit increase with increasing the refractive index of the sensing medium.

## 1. Introduction

Surface plasmon polaritons (SPPs) are electromagnetic waves, in the infrared or visible frequency-range, simultaneously propagating in the two adjacent media of a dielectric–metal interface, producing collective oscillations of the electronic plasma on the metal surface and consequently dissipation of electromagnetic energy. A very detailed discussion of electromagnetic modes propagating at the boundaries of a thin metallic film with two semi-infinite dielectric media at both sides of the film is given by J.J. Burke et al. [1]; book by S.A. Maier [2] is also an excellent reference to review dispersion relations and conditions which have to be satisfied by the refractive indices of the media, as well as, to review experimental techniques of excitation and detection of SPPs. 

A very common experimental set up for producing and detecting SPPs was first suggested by Kretschmann [3] in 1971. In this technique, the very well known experiment of total reflection is slightly modified by placing a thin film of a noble metal (gold or silver) on the flat surface of a D-shaped prism; SPPs are excited by evanescent waves traveling at the interface when angles of incidence are greater than the critical angle. The technique is also called attenuated total reflectance (ATR). SPPs are detected as a dip in the ATR spectrum (reflectivity vs. angle-of-incidence data). SPPs in the Kretschmann configuration started to be used experimentally as gas and biomolecular sensors since the early eighties; in the 90s, these sensors became commercial [4,5,6]. Other techniques (Otto configuration [7], grating coupling [8] and end fire excitation [9]) are also available for exciting and detecting SPPs. Typical ATR spectra [10,11] for plasmon absorption in noble metals show an absorption dip, with full width at half maximum (FWHM) of approximately 2°. On the other hand, it is very well known in the electromagnetic theory that an L/H/L structure of dielectric layers, where L and H, respectively, mean low and high refractive index, constitute a planar waveguide (PWG) structure. When a PWG structure over coats the noble-metal film of a plasmonic device, important variations in the ATR spectra occur [12,13,14,15,16,17], specifically, it results in a very sharp reflectance peak within a broad absorption dip. This effect is called plasmon induced transparency (PIT) and it has been named after the well-known phenomenon of electromagnetically induced transparency (EIT) [18,19,20], in which a narrow transparent window appears within a broad absorption band. Both effects, EIT and PIT are explained due to the interaction between a discrete level and a continuum of quantum states. Typically this interaction produces also an asymmetry in the resonance line shape; this was first explained by U Fano [21] and the phenomenon is also known as Fano resonance.

The state of the art for SPR coupled PWG sensors is that there are several publications [22,23,24,25,26,27] proposing theoretical designs, but there are only some few reports of experimental results; experimental studies published at the present are based on plasmonic metals like Ag, Al, Au and one based on Ge-doped SiO_2_ [15,28,29,30]. In this paper we report, firstly, the experimental observation of SPPs in the Kretschmann configuration produced at a TiN/air interface. For that, a thin film of titanium nitride was deposited on the flat surface of a D-shaped sapphire prism by dc magnetron sputtering; titanium nitride films have an optical appearance very similar to that of gold films; in addition, titanium nitride shows some properties such as: high hardness, high resistance to scratches and the ease of being produced by PVD techniques, which could be important for some specific applications. Our results were compared with those of pioneer studies of the plasmonic properties of titanium nitride [31,32,33,34,35,36,37]. Secondly, we report the construction of high sensitivity biosensors consisting of a layer of titanium nitride deposited on the flat surface of a D-shaped sapphire prism and over coated with two oxide layers: first a silicon dioxide layer or L-layer with n_L_ = 1.5 and then a niobium pentoxide layer or H-layer with n_H_ = 2.2. Then we got the H/L/TiN/Prism structure. Finally, we report and discuss the ATR angular responses in the Kretschmann configuration when the H-layer comes into contact with a sensing medium with refractive index n_S_ (n_S_ < n_H_), that is, an S/H/L/TiN/Prism structure. A narrow peak of the reflection intensity was observed within a broad absorption band and it was attributed to the plasmon induced transparency, or Fano resonance, due to the coupling between the continuum of quantum states of the surface plasmon polaritons at the TiN/SiO_2_ interface with the discrete level of a propagation mode of the plane waveguide structure constituted by: the silicon dioxide layer, niobium pentoxide layer and the sample or sensing medium (S). Sample media investigated were three: air, water and dried egg white. Each experimental spectrum was fitted to a theoretical one, by using the transfer-matrix method. From the fitted spectra we calculated the angular sensitivity and the figure of merit (FOM) for some reported sensors.

## 2. Materials and Methods 

Titanium nitride thin films were produced by dc magnetron sputtering onto the flat surface of D-shaped prisms made of sapphire. The target was a 3 inches diameter and 4 mm-thick disk of metallic titanium, 99.99% pure. The process was achieved in an atmosphere of 92% argon + 8% nitrogen. The plasma current was 400 mA and the deposition rate 20 nm per minute. The substrate temperature during the deposition process was 400 °C. Samples so produced were ready for ATR investigation in the Kretschmann configuration. A second type of samples were prepared by over coating the titanium nitride film with a layer of silicon dioxide and one of niobium pentoxide. For SiO_2_ sputter deposition, the target was a silicon disk, 3 inches diameter, 99.99% purity and 4 mm thick; the deposition atmosphere was 6% O_2_ + 94% Ar and the deposition rate was one micron per hour. Deposition conditions for Nb_2_O_5_ films were similar to those of silicon dioxide but a metallic target of niobium 99.99% pure was used.

X-ray diffraction was used for the verification of the crystalline structure of materials and scanning electron microscopy for verification of the multilayer structure and thicknesses measurements. The instruments were Bruker D8 and Hitachi 8230, respectively. X-ray diffraction was achieved in the theta-2theta configuration with cooper radiation of wavelength 0.154 nm.

Figure 1 shows the diffraction pattern of an Nb_2_O_5_/SiO_2_/TiN/sapphire structure. Typical diffraction peak at 36.70° was observed, which verified the structure of titanium nitride; rock salt structure with lattice parameter *a* = 0.42 nm. No diffraction peaks for oxide layers told us that they were amorphous; Figure 2 shows a SEM micrography where it can be seen the thickness of the titanium nitride thin film, as well as, that of silicon dioxide and niobium pentoxide layers. 

The experimental set up for measurements of reflectance vs. angle of incidence in the Kretschmann configuration is schematically shown in Figure 3. The source was a helium-neon laser of 632.8 nm, 1 mW. A polarizer and a thin slit were placed between the laser and the prism in such a way that p-polarized light hits on the prism; the rectangular light spot on the cylindrical surface of the prism was approximately 0.1 × 1 mm^2^. The intensity of the reflected beam was measured by using a photo detector constituted by a silicon photo diode and an amplifier. The device (TiN/sapphire prism or L/H/TiN/sapphire prism) and the detector were placed on two coaxial rotary tables in a theta-2theta system, in such a way that reflectance could be measured from 20 to 85° every 0.01°. Data were automatically recorded by using a Leybold Mobile-Cassy 2 computer interface.

In Figure 4, we schematically show the two types of devices, which were built and studied in this paper, the diameter of the D-prism was 25 mm; the glass cell was used, in this work, only for liquid media, i.e., water; fresh egg white was manually smeared on the Nb_2_O_5_ layer and waited 24 hours for drying at atmospheric conditions; measurements with air as sensing medium were accomplished with and without cell, the results were the same. 

From a theoretical point of view, the problem of calculating the reflectance vs. angle-of-incidence response in terms of the refractive indices (n_j_) and thickness of the different layers (t_j_) is completely solved by the electromagnetic theory [38,39]. In the transfer-matrix method, one 2 × 2 matrix is constructed, which relates the incident and reflected waves in medium 1 (the prism) with the outgoing ray in the last medium (or sensing medium); this matrix is expressed in terms of the incident angle, the refractive indices of all the involved media and the thickness of all the layers. Following these ideas, we made a program in JavaScript language, which simulates an ATR angle-spectrum for each set of n_j_ and t_j_ parameters. This program also works inversely: when we have an experimental ATR spectrum we can obtain the optical parameters (complex refractive index, in general, ñ_j_ = n_j_+ik_j_) and thicknesses t_j_ of the media, which better fit the experimental data. The fitting ideas have been proposed before for Zdzislaw et al. [6], as well as, Hibbins et al. [23] as methods to obtain spectral dependence of optical constants of the conducting layer and/or of a dielectric layer on top of it. In addition, several authors [12,13,14,15,16,17] have used simulation programs for the theoretical analysis of proposed new plasmonic structures. In this work, we used the program in both senses: simulation before manufacturing (designing) the plasmonic structures and fitting after the measurements of reflectance vs. angle of incidence. 

The parameters angular sensitivity, intensity sensitivity and figure of merit are defined [12,40], respectively, as:(1)$$S_{\theta }=\;\frac{\Delta \theta }{\Delta n}$$
(2)$$S_{I}=\;\frac{\Delta R}{\Delta n}$$
(3)$$FOM=\;\frac{\Delta R_{\max}}{\Delta n}$$

We have used angular sensitivity and figure of merit to characterize some of our sensors.

## 3. Results and Discussion

In Section 3 we firstly reported and discussed the experimental and fitted ATR spectra of SPPs at a TiN/air interface by using the Kretschmann configuration; from the fitted data we obtained the optical constants and complex electrical permittivity of titanium nitride at wavelength 633 nm. We also reported and discussed the experimental and fitted ATR spectra of the S/H/L/TiN/Prism multilayer structures, which were proposed here as prototypes of high sensitivity biosensors; from the data of fitted spectra, we obtained the angular sensitivity and the figure of merit for some sensors.

In Figure 5 we show an ATR spectrum for a basic device constituted by a thin film of titanium nitride deposited onto the flat surface of a sapphire D-shaped prism; this basic device is schematically shown in Figure 4a. Red line represents experimental data and the blue curve shows the respective fitting. The main feature of this angular spectrum was a very wide absorption band going from 36° (the sapphire–air critical angle) to 85°. The minimum reflectance was around 55°, the FWHM was approximately 40°. These features of our data agreed very well with those reported by N.C. Chen et al. [36]. Our experimental data were fitted by using the transfer-matrix method; the optical constants and thicknesses that better fit the data are shown in Table 1. From n and k data for TiN, the real and imaginary parts of relative dielectric permittivity were calculated and we obtained the value:(4)$$\varepsilon_{r}=\varepsilon_{1}^{\prime }+i\varepsilon_{2}^{\prime \prime }=-3.39+6.22i$$

Several authors [6,33,34,37] have reported theoretical and experimental data on relative dielectric permittivity and refractive index of titanium nitride prepared at different conditions; the real part of dielectric permittivity for TiN at wavelength 633 nm was in the range between −2 and −7 and the imaginary part was in the range between 1 and 7. In summary, we could say that optical constants and dielectric permittivity for sputtered titanium nitride films depended strongly on the sputter deposition conditions, there exists in the literature a range of values compatible with excitation of surface plasmon on TiN films and our results were inside this range.

It is also important to compare basic plasmonic devices based on a TiN film with conventional sensors based on a gold thin film; in ATR experiments and simulations reported by other authors [5,6,10,11], the absorption dip for SPPs at a gold/air interface has typically a FWHM around 2°; the big difference respect to SPPs at a TiN/air interface (40°) can be understood if we explore what happens with the simulated spectra when all parameters are fixed, except the k parameter. What actually happens is the broadening of the absorption dip when k decreases. For gold, the parameter k value is between 3.1 and 3.5 [11,12,16], whereas for titanium nitride produced in this work the k parameter values was 2.29 as shown in Table 1. Other interesting explanation of the differences in the ATR spectra at gold/air and TiN/air interfaces is given by Chen et al. [27]; they attribute the broadening of the band to the low mobility of electrons in titanium nitride associated to the larger damping of carriers. Both explanations, ours here and that of Chen et al., are connected through the basic relations between complex refractive index ñ, complex relative dielectric–permittivity $\varepsilon_{r}$ and complex electrical conductivity σ, described by Equations (5) and (6). Low mobility of carriers in titanium nitride was evident since carrier concentrations for gold and TiN were very similar (6 × 10^22^ cm^−3^), whereas, electrical conductivity of gold was 38 times larger than that of TiN. From the point of view of applications to sensors or biosensors, monolayer devices based on noble metals were more convenient because they presented much more resolution than those based on titanium nitride, because of the sharpness of the absorption dips. However, the situation was different with multilayer sensors based on titanium nitride and PWG structures, as we will describe next here.


(5)$$ñ\;=\;n+ik=\sqrt{\varepsilon_{r}}$$
(6)$$\varepsilon_{r}=\varepsilon_{r}^{\prime }+i\varepsilon_{r}^{\prime \prime }=\left(1-\frac{\sigma_{}\tau }{\varepsilon_{0}(1+\omega^{2}\tau^{2})}\right)-i\frac{\sigma_{}}{\varepsilon_{0}\omega (1+\omega^{2}\tau^{2})}\;$$


In Figure 6 we present nine ATR spectra for three sensors with the structure: S/Nb_2_O_5_/SiO_2_/TiN/Prism, S being the sensing medium: air, water or dried egg white. A scheme of this kind of device is in Figure 4b. The devices were constituted by a 33 nm thick titanium nitride film, the L-layer of SiO_2_ was 399 nm thick and the H-layer of niobium pentoxide was 102, 127 or 139 nm thick. Sensors were identified by the thickness of the layer with higher refractive index, i.e., the H-layer. The fitting parameters are giving in Table 2.

A first common feature of the nine ATR spectra in Figure 6 was the presence of a broad absorption band starting at an angle of incidence of approximately 56° and extending until 85°; this absorption band was explained as due to the surface plasmon polaritons set at the TiN/SiO_2_ interface. The second and most important feature of these spectra was the presence of a relatively narrow reflectance maximum within this broad absorption band; the physics explanation of this reflectance peak was the interaction between SPPs at TiN surface with the planar waveguide in the L/H/L structure, phenomenon known as plasmon induced transparency or Fano resonance. The angular position, the intensity and the width of the reflectance peaks varied with the thickness of the Nb_2_O_5_ layer, as well as, with the refractive index of the sensing media; increasing of the thickness of the H-layer shifts the angular position to higher values; greater values of the refractive index produce Fano resonances at larger angles of incidence. The matching between the fitting and the experimental curves was excellent, except at Fano resonances, in the cases of water and dried egg white; the reason was that the theoretical reflectance peak was extremely sharp (FWHM = 0.1°), and it was difficult to reach it experimentally with a laser spot of finite width (0.1 mm or 0.23°); nonetheless, the angular position of the reflectance maxima in the experimental and the fitted ATR spectra agreed very well, as shown in Figure 7; then experimental and theoretical angular sensitivities agreed very well, too. In Figure 8, we show a simulation of four spectra with refractive index very close to those of water and dried egg white; in Figure 8a for refractive indices 1.30 and 1.31; in Figure 8b for refractive index 1.550 and 1.551; these spectra were made with the fitting parameters for spectra shown in Figure 6c. From spectra in Figure 8 we obtained the angular sensitivity for samples with refractive index close to water
(7)$$s_{\theta }\;=\;30\;\deg\;RIU^{-1}$$
and for samples with refractive index close to egg white
(8)$$s_{\theta }\;=\;100\;\deg\;RIU^{-1}$$

These values were in good agreement with those obtained by Hayashi [12]; from Figure 8 we also got the figures of merit resulting FOM = 40 RIU^−1^ for samples with a refractive index close to water and FOM = 500 RIU^−1^ for samples with a refractive index close to dried egg white. It is clear that sensitivity parameters were better for samples with refractive index near dried egg white than water; we have to say, however, that changing the thickness of the H-layer to t_H_ = 150 nm would improve the sensitivity parameters:(9)$$s_{\theta }\;=\;36\;\deg\;RIU^{-1}$$
(10)$$FOM\;=\;50RIU^{-1}$$

On the other hand, it is possible to improve much more the figure of merit for samples with refractive index near water by using a material with a lower refractive index as the L-layer; for example, in Figure 9 we show a theoretical ATR spectra for n_L_= 1.35 and the other parameters the same as in Figure 6c. The result is
(11)$$s_{\theta }\;=\;30\;\deg\;RIU^{-1}$$
(12)$$FOM\;=\;450RIU^{-1}$$

## 4. Conclusions

Three prototypes of high sensitive biosensors were built by sputtering H/L/TiN multilayer structures on the flat surface of sapphire D-shaped prisms; the titanium nitride layer was approximately 33 nm thick, the L-layer was a 399 nm thick SiO_2_ film and the H-layer was a Nb_2_O_5_ film with thickness 102, 127 or 139 nm; the sensors were proved with three sensing media: air (n = 1), water (n = 1.33) and dried egg white (n = 1.55). For a specific sensor, the refractive index value of each sensing medium was identified with the angle of incidence value at which a Fano resonance occurred. The angular position of the Fano resonance increased for a larger thickness of the H-layer of the sensor and also for higher values of the refractive index of the sensing medium. Fano resonances were sharper (FWHM was lower) when they occurred at a greater angle of incidence, consequently, the angular sensitivity as well as the figure of merit were higher for Fano resonances happening between 70 and 80°. For sensors built and described in this paper, sensitivity parameters were better for samples with refractive index near egg white (n = 1.55) than near water (n = 1.33); however, based on theoretical calculations we have shown that it was possible to improve sensitivity parameters for samples with refractive index near water by increasing the thickness of the Nb_2_O_5_ layer and/or decreasing the refractive index of the L-layer. The great broadness of the plasmonic absorption-band (55–85°) allowed the Fano resonances to occur also in a great interval of angles of incidence, consequently, these sensors can be used for sensing materials in a wide interval of refractive index values, i.e., between 1.0 and 1.6; in particular, we hope these sensors could be useful for the study of molecular interactions involving streptavidin and biotin.

## Figures and Tables

**Figure 1 sensors-20-01784-f001:**
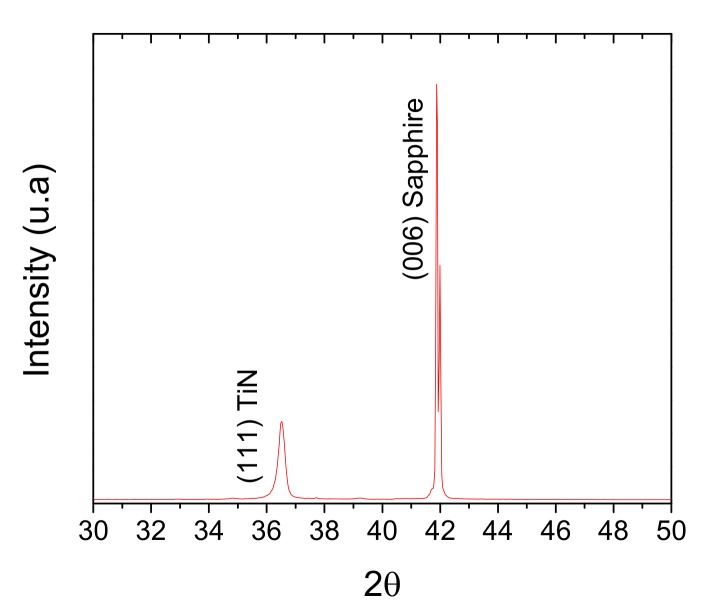
X-ray diffractogram of an Nb_2_O_5_/SiO_2_/TiN structure deposited on a sapphire plate.

**Figure 2 sensors-20-01784-f002:**
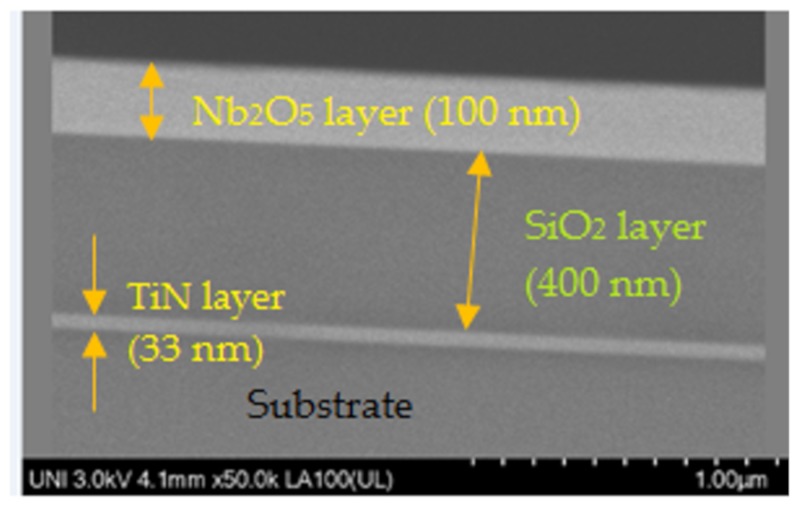
SEM micrography showing the cross-section of an Nb_2_O_5_/SiO_2_/TiN structure; the thicknesses were 33 nm for the TiN film, 400 nm for the SiO_2_ layer and 100 nm for the Nb_2_O_5_ layer.

**Figure 3 sensors-20-01784-f003:**
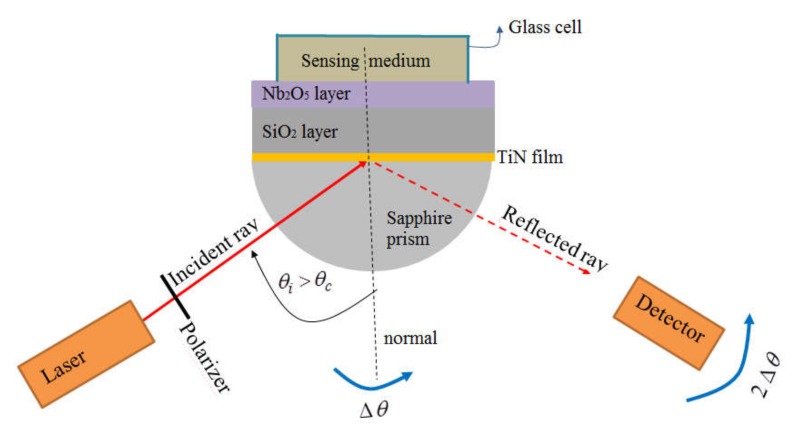
Schematic of the experimental setup for Reflectance vs. Angle-of-incidence measurements in the Kretschmann configuration; sensor and detector are mounted on two coaxial rotatory tables rotating in the plane of the figure, keeping the configuration theta-2theta; the incident ray is fixed respect to the laboratory.

**Figure 4 sensors-20-01784-f004:**
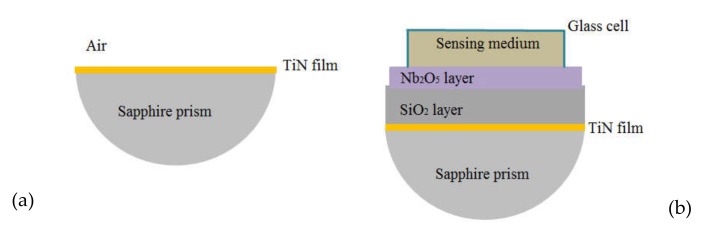
Schematics of (**a**) D-shaped sapphire prism coated with a thin film of titanium nitride and (**b**) prototype of a high sensitivity sensor.

**Figure 5 sensors-20-01784-f005:**
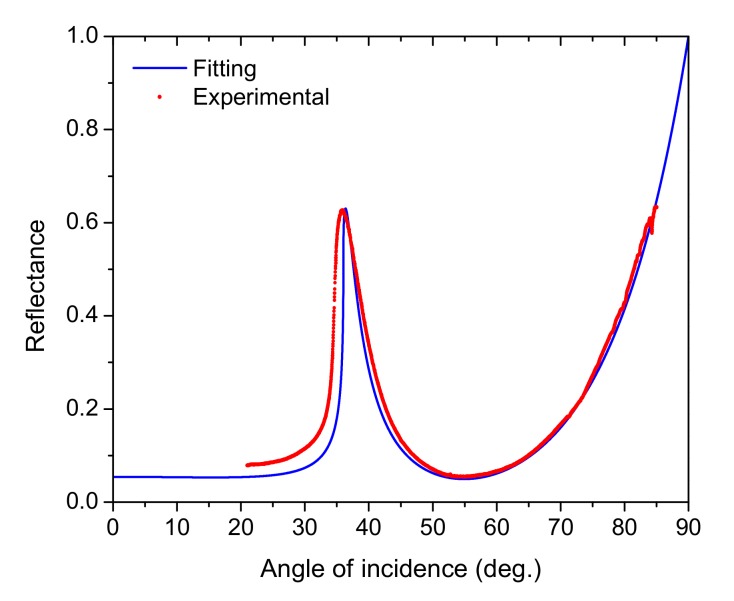
Experimental and fitted spectra for surface plasmon polaritons (SPPs) at an air/TiN interface, by using a sapphire D-shaped prism as substrate.

**Figure 6 sensors-20-01784-f006:**
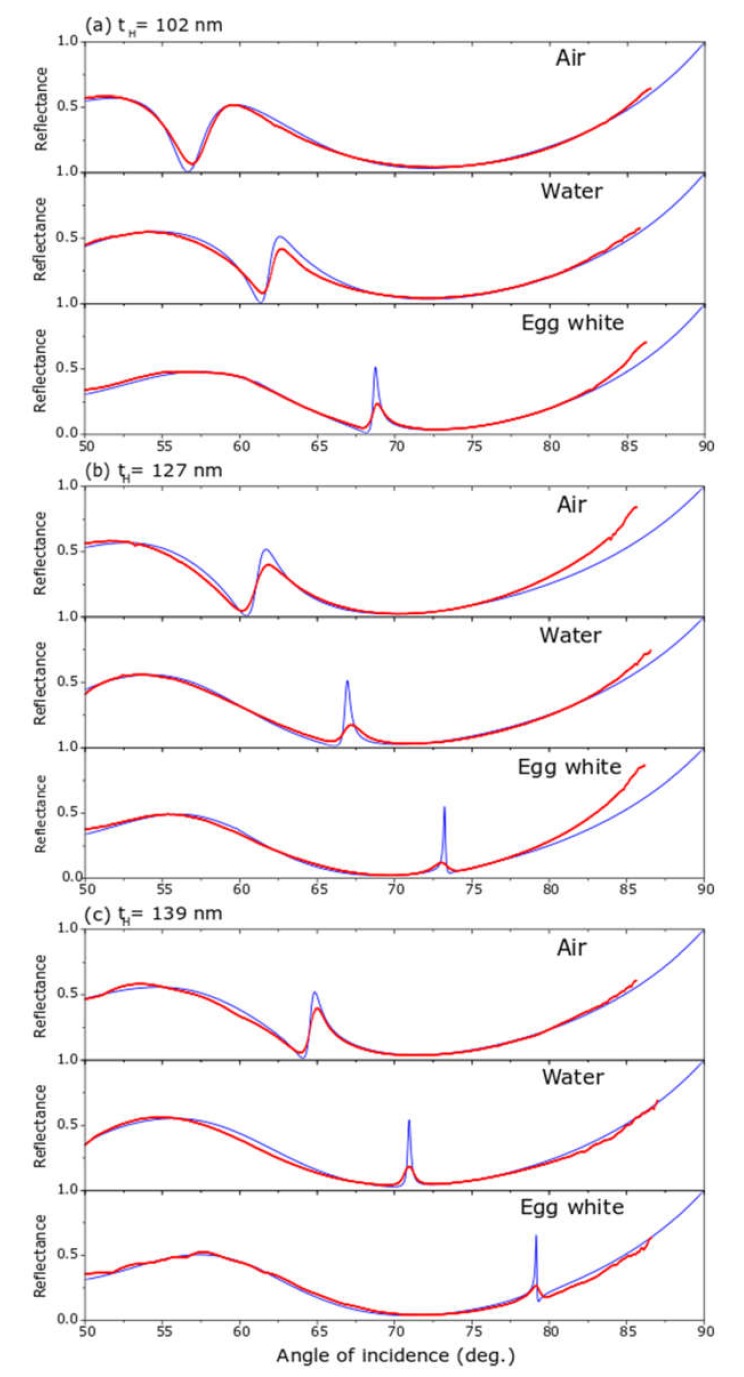
ATR spectra of three multilayer sensors for three sensing media: air, water and dried egg white. Sensors are identified by the thickness of the H-layer: (**a**) 102 nm, (**b**) 127 nm and (**c**) 139 nm. Red curves represent experimental data and the blue are the respective fittings.

**Figure 7 sensors-20-01784-f007:**
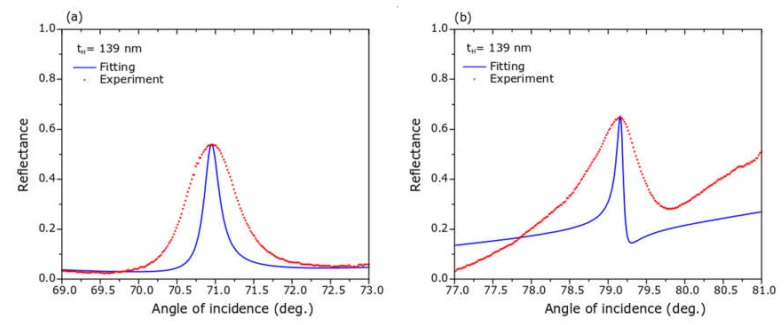
Partial sections of experimental and fitted ATR spectra shown in Figure 6c for water (**a**) and dried egg white (**b**). Data in these figures is the same as that in Figure 6c, except that the experimental data was multiplied by a constant factor. The purpose is the comparison of the angular position of theoretical and experimental reflectance peaks.

**Figure 8 sensors-20-01784-f008:**
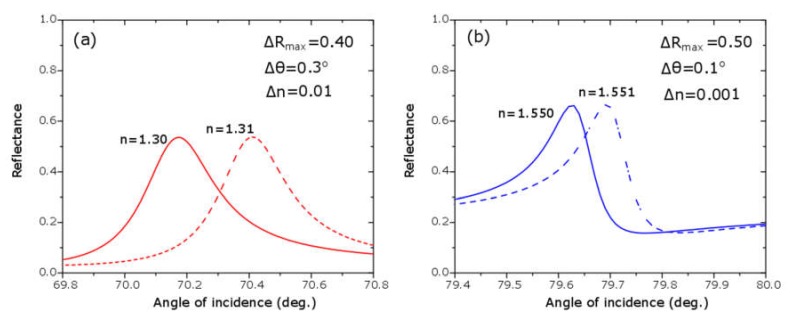
Fitted ATR spectrum 6c (t_H_ = 139 nm) near Fano resonances for samples with refractive index (**a**) near water and (**b**) near egg white, which allows one to estimate the angular sensitivity and the figure of merit for this sensor. For water: S_θ_ = 30 deg *RIU^−1^* and *FOM* = 40 *RIU^−1^*. For dried egg white: S_θ_ = 100 deg *RIU^−1^* and *FOM* = 500 *RIU^−1^*.

**Figure 9 sensors-20-01784-f009:**
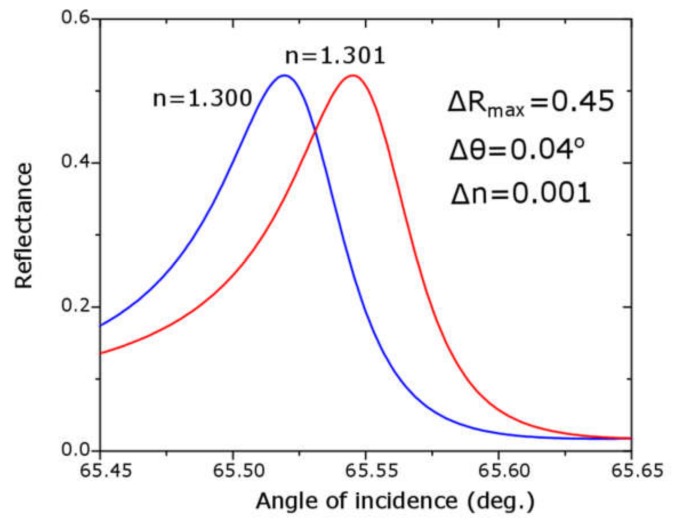
Theoretical ATR spectrum (t_H_ = 139 nm) near Fano resonances for samples with refractive index near water, which shows the possibility to improve sensitivity parameters by changing the refractive index of the L-layer. By using n_L_ = 1.35 and the other parameters equal to those of spectra in Figure 6c, it is possible to obtain for sensing media with a refractive index near water: S_θ_ = 30 deg *RIU^−1^* and *FOM* = 450 *RIU^−1^*.

**Table 1 sensors-20-01784-t001:** Thickness and optical parameters of the TiN thin film, which better fit the attenuated total reflectance (ATR) spectrum of Figure 5.

Material	Thickness	n	k
Air	Semi infinite	1.0	0
Titanium nitride	24 nm	1.36	2.29
Sapphire	Semi infinite	1.77	0

**Table 2 sensors-20-01784-t002:** Optical parameters, which better fix the ATR spectra of Figure 6.

Sensor	Material	Thickness (nm)	n	k
A	TiN	33	1.22	2.67
A	SiO_2_	399	1.49	0
A	Nb_2_0_5_	102	2.2	0
A	Air	semi infinite	1	0
A	Water	semi infinite	1.33	0
A	dried egg white	semi infinite	1.55	0
B	Material	thickness	n	k
B	TiN	33	1.22	2.67
B	SiO_2_	399	1.49	0
B	Nb_2_0_5_	127	2.2	0
B	Air	semi infinite	1	0
B	Water	semi infinite	1.33	0
B	dried egg white	semi infinite	1.55	0
C	Material	thickness	n	k
C	TiN	33	1.22	2.67
C	SiO2	399	1.49	0
C	Nb205	139	2.2	0
C	Air	semi infinite	1	0
C	Water	semi infinite	1.3317	0
C	dried egg white	semi infinite	1.55	0

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
