# Peer review of "High Sensitive Biosensors Based on the Coupling Between Surface Plasmon Polaritons on Titanium Nitride and a Planar Waveguide Mode"

_sensors, 2020, doi:10.3390/s20061784_

Round 1

Reviewer 1 Report

Very clear and solid paper. Relevance of the topic without a doubt. But there are a few ambiguities.

Firstly, in the strings 213-214 the mismatch of the fitting and experimental curves is explained by lack of angular resolution while but distance between the experimental points much less than this limit and they are clearly show smooth dependance, not very noisy. Besides, if the constructed experimental setup can not resolve FWHM=0.1deg, how it can resolve, for example, two curves shown in fig. 8b?

Secondly, in the strings 224-225 it is mentioned that sensitivity parameters are better for samples with refractive index near dried egg white than water, but the issue of dependance of sensitivity on refractive index of sensing medium is not fully disclosed. Does it mean that the higher refractive index of sensing media the higher angular sensitivity? Or it is cirrect only within certain range of refractive indices?

Author Response

Answer 1: There are two experimental aspects which contribute to the angular resolution of our measurements. (1) the angular speed of the rotary tables and (2) the width of the laser beam. The angular speed of our rotary tables is quite good, good enough. The width of the laser spot is normally good to follow the theoretical curve (fitting), except when the teoretical curve R vs angle of incidence changes vary rapidly with the incidence angle, i.e., just when Fano resonance happens.  In figura 8b we show theoretical curves obtained with the experimental parameters of all the layers, according to the shown fitting.

Answer 2: It is a kind of rule of thumb obtained from our experimental data. For example for sensor with tH= 127 nm we can see that for n = 1 (air) FWHM = 5o, for n = 1.33 (water) FWHM = 2and for n = 1.55 (egg white) FWHM < 1o. Finally, sensitivity parameters are better for sharper resonances.

Reviewer 2 Report

Comments for authors:

The submitted manuscript experimentally studies sensing ability of a multilayer structure deposited on a prism, which is a famous Kretschmann configuration. The sensing principle is based on excitation of SPP mode and waveguide mode in the high index layer, and coupling between the two modes leads to PIT or Fano resonances. One novel point of the manuscript is by replacing conventional noble metal with titanium nitride, showing some specific advantages, such as high hardness and ease of being produced etc. The reported best angular sensitivity is 100 deg/RIU and FOM (maximum change of reflectance versus index change) is 500 /RIU. However, such sensitivity and FOM are not in a high level, compared to previous SPP sensors (such as in Laser Photonics Rev. 5, 571 (2001)). Overally, the present manuscript does not include real improvement in sensing behavior and the physics behind the sensors have been well known before (such as in Opt. Express 24, 20080 (2016)). So the novelty is lacking. As an experimental work, the measurement datas are scarce (only in Fig. 6) and the angular sensitivity and FOM are only derived from the theoretical fitting curves. If more sensing measurement is presented and the sensitivity and FOM are derived from the experimental curves, that will be much more convincing and attractive. In addition, the organization of the manuscript is not so well. Some suggestions are listed as follows:

  1. The arrows in Fig. 2 are not placed correctly.
  2. It is not necessary to present Fig. 4 solely. The Fig. 4(a) and Fig. 4(b) can be inserted into Fig. 5 and Fig. 6, respectively.
  3. Equation (7) to (12) is actually 6 values (angular sensitivity and FOM). These should not be listed as equations.
  4. Format of the references should be consistent (such as Ref. 13 and Ref. 15 are different from Ref. 3 and Ref. 4). Please double check.

Therefore, in my opinion, the manuscript should be rejected at present.

Author Response

A) Answer to the general comment:

Reviewer 2 first accepts that the novelty of this paper is the use of titanium nitride instead of gold or silver, then he realizes that sensitivity parameters are not better than those for conventional devices and finally he concludes that there is not novelty. I feel that the reviewer is mixing concepts novelty of the paper with quality of the devices. We recognize that this paper does not contribute to the physics of multilayer devices. The aim of this work, however,  is more modest, just to show to biologists and scientists working in the sensors area the possibility to use sensors with  new structures based on other materials than gold or silver. We recognized (it is writen in the paper) that measurements can be improved by thining the diameter of the laser beam and using other finer detection techniques than  just a biased photodiode.

B) Answer to the listed comments:

1) Thanks, we agree that figure 3 can be improved. It was done.

2) Yes, what reviewer 2 says is a good alternative, however figure 6 has already too much information.

3) Equations 1 to 3 shoud not be listed. Again a good alternative but I feel it would unnecesarily complicate the already done structure of the paper.

4) Thanks for this important comment. We have edited the references according to the format of the Sensors journal

Reviewer 3 Report

This manuscript presents high sensitivity biosensors that work based on the coupling between Surface Plasmon Polaritons and a planar waveguide mode. 

  1. The novelty of the work is vague. The reader must understand the novelty of the work by reading the abstract. I expect the authors to clarify and highlight the novelty if the manuscript goes for revision.
  2. The manuscript has been well organized, but it seems like they have done something that already has been done before and their contribution is not in a way that could be published as a journal paper. Please express the exact contribution of this manuscript compared to the literature.
  3. The manuscript has grammatical mistakes and punctuation errors. They need to be fixed.
  4. Although the mechanism of the sensing is obvious from Figs. 6 and 7 (from reflectance vs angle of incidence), this reviewer still thinks that there is still a need for more analysis and explanation. 
  5. Can the authors explain the potential application for their proposed biosensor?

Author Response

Answers to comments 1 and 2: Novelty

In the eighties when photovoltaic cells were developped thousands of papers were written about polycristalline silicon and amorphous silicon.

Up to now we have not seen any paper reporting Fano resonance in multilayer structures containing titanium nitride as plasmonic material.

Any way, thanks for your kind comments which allows us to improve our manuscript by adding three small paragraphs: one in the abstract, one in the introducion and one in the conclusions. We also introduce 9 new references to show that  most of the papers about SPR coupled WG sensors up to now are theoretical proposals than experimental devices. There exist, of course, some few experimanteal devices based on metal as plasmonic material but not on titanium nitride. We explained also that the importance of TiN as plasmonic material is the great broadness of the plasmonic absorption-band which allows to sense materials in a greater range of refractive index values.

Answer to comment 3:

Perhaps there are some grammatical erros. I will appreciate very much the editors of the journal to postpone that correction.

Answer to comment 4 more analysis and explanation

The purpose of the proposed paper is to report the experimental results and as a waranty that  the experiments were well done we show also the theoretical fitting.

 Answer to comment 5

At this moment, we have started a MSc. thesis to investigate the applications of the sensors described here in the study of the interaction between biotinylated molecules with streptavidin.

Round 2

Reviewer 2 Report

Comments for authors:

The authors show great effort to improve the quality of the manuscript. The potential application of the proposed results in sensing biological materials, such as biotin and streptavidin, is proclaimed clearly, which is encouraging for me. As an experimental work in SPR sensing combined with Fano resonances, it might attract researchers’ interests and give some useful information for further improving the SPR sensing ability and application scope. Additionally, the results in the manuscript would draw subsequent scientific researches in the near future.

In a word, the revision is fine and it can be accepted for Sensors.

Author Response

It is not ncesary to reply. The referee approved the manuscript.

Reviewer 3 Report

The authors have answered the comment raised by the reviewers and I think the current version of the paper deserves publication.

Author Response

(The authors gave the same response as above.)
